# Integrating status-neutral and targeted HIV testing in Zimbabwe: A complementary strategy

Hamufare D. Mugauri[1]*, Owen Mugurungi[2], Joconiah Chirenda[1], Kudakwashe Takarinda[3], Prosper Mangwiro[4], Mufuta Tshimanga[1]

**1** Faculty of Medicine and Health Sciences, The University of Zimbabwe, Global, Public Health and Family Medicine Department, Harare, Zimbabwe, **2** Ministry of Health and Childcare, AIDS and TB Unit, Harare, Zimbabwe, **3** Organisation for Public Health Interventions and Development (OPHID), Harare, Zimbabwe, **4** Independent Consultant, Zimbabwe

\* dumiwaboka@gmail.com

## Abstract

### Introduction

Zimbabwe exclusively implemented targeted HIV testing until 2022 when Status-neutral testing was embraced. Whilst targeted testing aims to expand access and uptake of testing among high-risk individuals, status-neutral testing emphasizes post-test linkage to prevention and treatment services. To address how the two concepts relate in practice, we explored how status-neutral and targeted testing concepts correlate, in developing a double-edged strategy for effective case identification and linkage to prevention and treatment.

### Methods

We conducted a cross-sectional study on 36 multi-stage sampled sites across 4/10 provinces of Zimbabwe. A national screening algorithm was used to determine patient risk profiling and eligibility for testing. Screened-out patients were offered HIVST. Both screened and non-screened patients were tested and analysed for positivity ratios and linkage to post-test services. Epicollect5 was used to collect data and analysed using EpiData software and Stata. Univariate, bivariate and multivariate analyses were conducted at a 5% significance level.

### Results

Of 23,058 HIV tests done, females constituted 55% (n = 12,698), whilst 63.5% (n = 14,650) were retested. Through screening, at-risk patients contributed 75.1% to the overall positivity (1,296/1,727), from 66% (n = 15,289) of the total HIV tests conducted. All screened-out patients were non-reactive on HIVST: 1,182/1,182. The 45–49-year category was 3.6 times more likely to test positive (a95%CI:2.67,4.90). Males were 3.09 times more likely to test positive in adjusted analysis (a95%CI: 2.74, 3.49). First tests were 65% more likely to test HIV positive (a95%CI: 1.43, 1.91) whilst screened patients were 3.89 times more likely to link to HIV prevention services (a95%CI: 3.05, 4.97), against 25.5% (n = 1,871) linkage among patients not screened.

**Data availability statement:** All relevant data are within the manuscript and its Supporting Information files.

**Funding:** The author(s) received no specific funding for this work.

**Competing interests:** The authors have declared that no competing interests exist.

## Conclusion

The complementarity of the status-neutral and targeted testing approaches is evident from our results. By prioritizing high-risk individuals for testing and ensuring comprehensive linkage to both prevention and treatment services, these integrated strategies can effectively identify and manage people living with HIV. This combined approach optimizes resource use, particularly in low- and middle-income countries, and contributes to improved health outcomes and reduced HIV transmission rates.

## Introduction

Status-neutral HIV testing represents an innovative paradigm in HIV education, testing, and treatment, emphasizing a continuous care model irrespective of an individual's HIV status [1]. This approach ensures that all individuals, regardless of their HIV diagnosis, receive uniform treatment from the initiation of the testing process and are subsequently linked to appropriate services based on their test outcomes. Additionally, this model aims to enhance health outcomes, prevent new HIV infections, and ultimately create a scenario where HIV transmission is halted through universally accessible prevention and treatment strategies [2].

The concept prioritizes interventions tailored to the needs of populations at risk for or living with HIV, rather than segregating services into prevention or care categories [3]. The status-neutral approach to HIV prevention and care identifies the HIV testing moment as the critical entry point to care. At this juncture, clients' needs are comprehensively assessed, and they are subsequently engaged and connected to appropriate services based on these needs, irrespective of their HIV test results [4].

First introduced by the New York City Department of Health and Mental Hygiene in 2016, this paradigm represents a comprehensive prevention system encompassing all individuals affected by HIV, irrespective of their HIV status [5–7]. It delineates detailed steps to achieve an undetectable viral load and outlines strategies for effective combination HIV prevention. The concept is grounded in the premise that most countries have attained the 95% targets and should now focus on consolidating these achievements by equally addressing the needs of HIV-negative individuals, who were previously overlooked in the pursuit of positivity ratio targets [8].

The status-neutral concept is rapidly gaining global acceptance. Organizations such as the World Health Organization (WHO) and the Centres for Disease Control and Prevention (CDC) actively promote it through various initiatives [9]. The adoption of this concept varies among countries, influenced by their unique contexts and needs.

In Africa, particularly in sub-Saharan regions, the status-neutral approach has been integrated into existing HIV testing frameworks to enhance the effectiveness of HIV prevention and treatment programs. This approach ensures that HIV testing and related services are offered to all individuals, regardless of their HIV status, promoting continuous engagement in care. Countries like Zimbabwe have been at the forefront of implementing this approach, incorporating it alongside targeted testing methods to improve linkage to care and retention in treatment, thereby addressing both prevention and treatment comprehensively [7]. This dual strategy, which involves integrating the status-neutral approach with targeted testing methods, is yet to be fully understood, highlighting a significant knowledge gap. Examining this dual strategy is crucial because it has the potential to enhance the effectiveness of HIV prevention and treatment programs by ensuring continuous engagement in care regardless of HIV status, while also targeting high-risk populations for more focused interventions.

Understanding how these two approaches can complement each other will contribute to more effective and inclusive healthcare strategies, ultimately improving health outcomes in sub-Saharan Africa. Our work aims to fill this gap by providing comprehensive insights into the integration and impact of these strategies, positioning itself as a valuable resource for policy-makers and healthcare providers.

Zimbabwe, located in sub–Saharan Africa, is one of the countries severely affected by the HIV pandemic, where HIV remains firmly established as a generalized epidemic with a prevalence of 11,01% and incidence of 17% translating to approximately 23,000 new infections every year [10,11] (Fig 1).

To address her predicament, Zimbabwe has been implementing a Targeted testing model that prioritizes high-risk individuals through a screening algorithm, adopting differentiated testing models that include strengthening Index testing- a proven high-yield HIV testing model [12].

Whilst, remarkably, the country has achieved the 95% targets, according to UNAIDS HIV estimates, the country still bears a generalized epidemic contributed by population and geographically varied sub-epidemics requiring innovations for case finding and effective HIV prevention packages [13].

This context demands that Zimbabwe, like her Southern African counterparts, needs interventions designed to arrest ongoing transmission of HIV, identify the remaining cases, and put them on effective life-long treatment to bring the pandemic to an end.

Zimbabwe presents a unique case study for several reasons. First, Zimbabwe has one of the highest HIV prevalence rates in sub-Saharan Africa, making it imperative to explore innovative strategies to curb the epidemic. Second, the country's healthcare infrastructure, while robust in many areas, faces significant resource constraints, necessitating the efficient use of available resources. Third, Zimbabwe's existing HIV testing programs provide a strong foundation upon which status-neutral strategies can be integrated, offering a practical context for assessing the effectiveness of combined approaches.

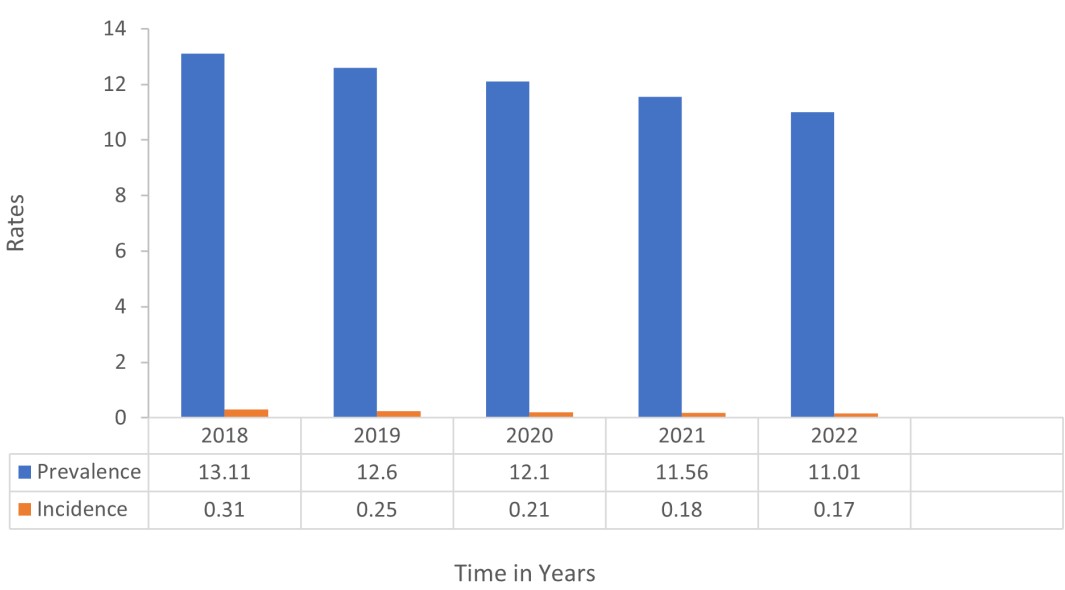

| | 2018 | 2019 | 2020 | 2021 | 2022 |
|---|---|---|---|---|---|
| ■ Prevalence | 13.11 | 12.6 | 12.1 | 11.56 | 11.01 |
| ■ Incidence | 0.31 | 0.25 | 0.21 | 0.18 | 0.17 |

Time in Years

■ Prevalence   ■ Incidence

**Fig 1. HIV Prevalence and Incidence Rates in Zimbabwe, 2018-2022 (Source: UNAIDS HIV Estimates).**

Moreover, Zimbabwe's diverse population and varying levels of access to healthcare services across urban and rural areas provide a comprehensive setting to evaluate the adaptability and scalability of the status-neutral approach. This makes Zimbabwe an ideal context for understanding how these strategies can be tailored and implemented in similar resource-limited settings across the region.

This paper thoroughly examined the complementariness of the status-neutral and targeted approaches to HIV testing in the context of a generalized epidemic, high new infections, and established ongoing transmission of HIV. By exploring how these strategies can work together to enhance HIV prevention and treatment efforts, we seek to provide insights and recommendations for the context-specific application of the status-neutral concept. Our goal was to highlight the potential benefits of integrating these approaches to improve healthcare outcomes in sub-Saharan Africa, particularly in countries like Zimbabwe.

## Materials and methods

This study employed a cross-sectional design to examine the integration of the status-neutral approach within existing HIV testing frameworks in Zimbabwe. Data were sourced from facility records as entered into the DHIS2 database. The cross-sectional nature of the study allowed us to capture a snapshot of the current state of HIV testing and treatment, providing valuable insights into the effectiveness and challenges of implementing the status-neutral approach alongside targeted testing methods.

### Setting

Zimbabwe is a landlocked, low-income country in Southern Africa located between Botswana, South Africa, Mozambique, and Zambia with an estimated population of 16,3 million and a human development index of 0.593, ranked number 174 globally out of 189 countries in 2022 [14,15]. The country is divided into two urban provinces, eight rural provinces and 62 districts.

The AIDS and TB Programme (ATP) is mandated to coordinate the development of HIV/AIDS health policies and set up national standards and guidelines as part of the national response to HIV in Zimbabwe. Four sub-units under ATP, namely, HIV Prevention, Care and Treatment, Prevention of Mother to Child Transmission (PMTCT) and Monitoring and Evaluation (M & E). These sub-units are delegated to ensure seamless yet specialised programming to ensure adequate response to the pandemic [16].

The HIV Prevention program oversees the activities of HIV Testing Services (HTS) activities with the ATP. Since 2016, the HTS programme has been pursuing targeted testing as an approach to reduce testing volumes, increase efficiency in HIV testing and enhance the identification of people living with HIV, to enrol them on life-long Antiretroviral Therapy (ART). Eligibility for HIV testing is done using a validated Screening algorithm [17]. In this algorithm, high-risk individuals are offered provider-delivered testing whilst those screened out are offered HIVST kits for self-screen. Following a negative test result, the patient is further screened for eligibility for combination prevention which includes PrEP.

**Specific study site.** The study sites included four provinces selected out of the ten provinces in Zimbabwe, based on their diverse geographical and demographic characteristics as well as varying levels of HIV prevalence and incidence. *Manicaland Province*: Located in eastern Zimbabwe, Manicaland is the second-most populous province after Harare, with a population of approximately 1.75 million as of the 2022 census. This province has been significantly impacted by the HIV epidemic, with a prevalence of 9.40% in 2024. This makes it a critical area for studying HIV interventions [18,19]. *Mashonaland West Province*:

Situated to the north of Zimbabwe, Mashonaland West shares an international border with Zambia. It borders several other provinces internally, including Midlands, Matabeleland North, Mashonaland Central, Harare, and Mashonaland East. The region had a notable HIV prevalence of 9.60% in 2024, warranting focused efforts on HIV testing and treatment [19,20]. *Matabeleland South Province:* This province covers the southeastern plateau of Zimbabwe and stretches to the borders with Botswana and South Africa. The area is characterized by high HIV incidence, particularly in cross-border communities, with a prevalence rate of 17.30% in 2024, highlighting the need for targeted HIV prevention and treatment strategies [19,21]. *Midlands Province*: Midlands province spans an area of 49,166 square kilometres and has a population of approximately 1.61 million. With its diverse population and significant HIV prevalence rate of 10.94% in 2024, this province provides a vital context for examining the effectiveness of HIV interventions [19,22].

These provinces were selected to provide a comprehensive understanding of the status-neutral approach and its integration into existing HIV testing frameworks across diverse settings. The varying HIV prevalence and incidence rates in these regions underscore the importance of context-specific strategies for HIV prevention and treatment.

## Sampling

Multi-stage sampling was done to randomly select 4 out of 10 Zimbabwe provinces using the lottery method. Further, 3 districts per province were randomly selected resulting in 12 districts. All health facilities across the 12 districts were included in the study, resulting in a total of 36 health facilities. The sampling criteria were designed to ensure a balanced representation, incorporating a mix of high and low-volume facilities, as well as urban and rural sites (Table 1).

## Client population

All clients who were tested for HIV and documented in HIV Testing registers, at the 36 sampled facilities between 1 October 2023 to 31 December 2023 and aged 15 years and above, were included in the study.

## Inclusion criteria

Individuals who underwent HIV testing within the specified study period.

**Table 1. Study sites, Zimbabwe, 2024.**

| Manicaland Province | Mash West Province | Matabeleland South Province | Midlands Province |
|---|---|---|---|
| Mutare District<br>Mutare Provincial Hospital<br>Zimunya Clinic<br>Mt Zuma Clinic | **Chegutu District**<br>Katanga Utano Clinic<br>Pfupajena Municipal Clinic Selous Clinic | **Gwanda District**<br>Gwanda Provincial Hospital<br>Phakama Clinic<br>Manama Mission Hospital | **Gweru District** Gweru Provincial Hospital<br>Chikwingwizha Mission Hospital<br>Lower Gweru Clinic |
| Chipinge District<br>Chikore Mission Hospital<br>Chipinge Town Clinic<br>Tanganda Rural Health Centre | **Makonde District**<br>Chinhoyi Provincial Hospital<br>Chikonohono Municipal Clinic<br>Alaska Municipal Clinic | **Mangwe District**<br>Plumtree District Hospital<br>Tshitshi Clinic<br>Dingumuzi Clinic | **Kwekwe District**<br>Kwekwe General Hospital<br>Amaveni Clinic<br>Nyoni Rural Health Centre |
| Makoni District<br>Rusape District Hospital<br>Katsenga Rural Health Centre | **Sanyati District**<br>Kadoma General Hospital<br>Ordoff Clinic<br>Waverly Municipal Clinic | **Umzingwane District**<br>Nhlangano Clinic<br>Nswazi Clinic<br>How Mine Clinic<br>Kumbudzi RHC | **Zvishavane District**<br>Mandava Health Centre<br>Mabasa Clinic<br>Mtambi Clinic |

Participants who provided sufficient demographic and clinical information necessary for analysis.

## Exclusion criteria

Individuals with incomplete or missing data on key variables.

Participants whose testing data were not available in the DHIS2 database.

## Sample size calculation

The sample size for this study was determined using statistical power analysis to ensure adequate power to detect significant differences between groups. The primary outcome measure was the effectiveness of the status-neutral approach in promoting linkage to care for both HIV-positive and HIV-negative individuals. This approach ensures that all individuals, regardless of their HIV status, are linked to appropriate health services. Using secondary data from existing HIV testing programs, our analysis aimed to evaluate how well the status-neutral approach facilitated continuous engagement in care for everyone tested, thereby improving overall health outcomes within the context of targeted HIV testing

The following assumptions and parameters were used

Effect Size: Based on previous studies, an effect size of 0.3 was considered clinically significant, Significance Level ($\alpha$): A two-tailed significance level of 0.05 was used.

Power ($1-\beta$): To achieve a power of 80%, the sample size was calculated to detect the specified effect size, Proportion of Positive Cases: An estimated proportion of 10% HIV-positive cases was assumed based on historical data.

The sample size was calculated using the formula for comparing two proportions and using the above parameters and formula, the required sample size was calculated to be approximately 385 participants per group. To account for potential dropouts and incomplete data, an additional 10% was added, resulting in a final sample size of 424 participants per group.

However, the actual data set included 23,058 individuals who were tested for HIV within the study period. The larger sample size was due to the inclusion of all data available from the DHIS2 database, providing a more comprehensive analysis. This substantial increase in the sample size enhances the study's statistical power and the generalizability of the findings, allowing for a more robust evaluation of the status-neutral approach's effectiveness.

## Data variables, sources of data and data collection

Data were extracted from District Health Information System version 2 (DHIS2) as an Excel report. The data was accessed on the 26th of February 2024 for research purposes. The data was exported to EpiData Analysis version 2.2.2.186 (EpiData Association, Odense, Denmark) and Stata v14 (Stata Corporation College Station, Texas, USA) for further cleaning and analysis. The following variables were collected: HTS number, name of the facility, age, sex, screening for an HIV test, reason for an HIV test, HIV test result, linkage to post-test services (yes/no), and specific services linked to. To ensure privacy, patient names were not included in the final dataset used for analysis. Therefore, no personal identifiers were collected in the analysed data.

*Variables Collected* Screening for an HIV test with This variable refers to the initial assessment process conducted to determine an individual's risk of HIV infection and the need for an HIV test. It includes questions about sexual behaviour, history of sexually transmitted infections, and other risk factors that might indicate the need for HIV testing. Linkage to post-test services: This variable indicates whether individuals who tested positive for HIV were connected to appropriate follow-up services. This includes referrals to healthcare facilities,

counselling services, and support groups to ensure continuous engagement in care and treatment adherence.

Specific services linked to: These variables detail the specific types of post-test services that individuals were connected to. These services may include antiretroviral therapy (ART), prevention of mother-to-child transmission (PMTCT) programs, pre-exposure prophylaxis (PrEP) for those at high risk, and other medical and social support services aimed at improving health outcomes for people living with HIV.

**Variables included in the adjusted models.** The adjusted models included the following variables based on their relevance and significance in the unadjusted analyses: *Age, Sex, and High-risk behaviour*: Individuals who reported behaviours such as multiple sexual partners, inconsistent condom use, or history of of of of of sexually transmitted infections, *Screening for HIV test*: The initial assessment process to determine the individual's risk of HIV infection, *Linkage to care and prevention*: Whether individuals were successfully linked to post-test services, such as antiretroviral therapy (ART) for HIV-positive individuals or pre-exposure prophylaxis (PrEP) and other preventive measures for HIV-negative individuals, *Testing district*: Geographic location where the testing took place.

While the primary analysis of risk profiles used log-binomial regression, the variables in Table 3 were analysed using Poisson regression with robust error variance to approximate prevalence odds ratios. This approach ensures the reliability and stability of the estimates, aligning with the study design.

## Analysis and statistics

Socio-demographic characteristics of participants were summarized using percentage for categorical data and mean (standard deviation) or median (interquartile range) for continuous data depending on whether they are normally distributed or not. The number and proportion with a 95% confidence interval were used to summarize all clients tested for HIV during the study period, the outcomes of the test and linkage to post-test services as documented in the respective registers.

To assess the risk profiles and estimate relative risks, we initially employed a log-binomial regression model. This model is appropriate for estimating relative risks directly when analysing binary outcomes. However, due to potential convergence issues often encountered with the log-binomial model, especially in cases of sparse data or rare outcomes, we also utilized Poisson regression with robust error variance. This method serves as an alternative approach, providing similar estimates to the log-binomial model and ensuring better convergence and stability in the results. Those variables with a p-value < 0.25 in the unadjusted analysis were included in adjusted models. The unadjusted and adjusted prevalence odds ratios at 5% significance levels (95%CI) were expressed as a measure of association.

The Poisson regression with robust error variance is widely used to approximate relative risks when the log-binomial model fails to converge. Consequently, while the primary analysis of risk profiles used the log-binomial regression, the variables in Table 3 were analysed using Poisson regression to ensure the reliability and stability of the estimates.

## Ethics approval

Our study was exempted from ethical clearance because it utilized routinely collected data which was extracted from the electronic database (DHIS2). The data we received were de-identified, with all patient-identifying information removed before analysis. This ensured the privacy and confidentiality of the individuals' information. As a result, no primary data

were collected, and the use of de-identified data qualified the study for ethical exemption. Despite this exemption, we recognized the sensitivity of HIV-related data and implemented stringent privacy safeguards to protect participants.

## Results

### Demographic characteristics

Out of the 23,058 patients tested for HIV, 55% (n = 12,698) were female. The majority of patients, 54.7% (n = 12,615), were within the age range of 25–49 years. Eligibility screening before testing was documented for 66.3% (n = 15,289) of the patients, while 33.7% were not screened. Among those tested, 63.5% (n = 14,650) underwent retesting, and 1,727 individuals tested positive, resulting in a positivity ratio of 7.5%. Across the districts, Gweru reported the highest number of tests conducted at 15.5% (n = 3,575), whereas Mangwe reported the lowest at 4.5% (n = 1,040) (Table 2).

### Screening, testing outcomes and post-test linkages

Out of the 23,058 patients who attended the sampled healthcare facilities, 66.3% (N = 15,289) were screened for eligibility before testing. Among these screened patients, the positivity ratio was 8.5% (N = 1,296), with nearly all (N = 1,294, 99.8%) being enrolled into care. The positivity ratio among screened patients accounted for 75.1% of the total positivity observed in this study (1,296/1,727). In contrast, among the 7,769 patients (33.7%) who were not screened before testing, 431 (5.5%) tested positive, representing 24.9% of the overall positivity (431/1,727). Of the 7,338 patients (94.5%) who tested negative, 1,871 (25.5%) were linked to HIV prevention services (Fig 2).

### HIV positivity and linkage

The overall positivity ratio observed in this study was 7.5% (1,727/23,058). Individuals in the 45–49-year age group were 3.6 times more likely to test positive for HIV (adjusted 95% CI: 2.67, 4.90). Males had a 3.09 times higher likelihood of testing HIV positive in the adjusted analysis (adjusted 95% CI: 2.74, 3.49), with a positivity ratio of 8% (n = 912). Initial tests were 65% more likely to yield a positive HIV result (adjusted 95% CI: 1.43, 1.91). Additionally, patients who were screened before testing were 3.89 times more likely to be linked to at least one HIV prevention service (adjusted 95% CI: 3.05, 4.97) (Table 3).

## Discussion

A key finding in this study is that the status-neutral approach to HIV testing complements targeted HIV testing by creating a more inclusive and comprehensive testing framework. This dual strategy allows for prioritized testing of high-risk populations while ensuring that all individuals, regardless of their HIV status, have access to HIV prevention and treatment services. By integrating these approaches, we can enhance case identification, improve linkage to care, and ultimately reduce HIV transmission rates. This model aligns with the Comprehensive 95% targets set by UNAIDS, which emphasizes the importance of linkage to comprehensive HIV prevention services. The analysis demonstrated that the algorithm used for identifying high-risk individuals for HIV testing was effective in prioritizing those most likely to benefit from testing and subsequent linkage to care. Our results showed that individuals identified through the algorithm had a higher likelihood of testing positive for HIV and being linked to appropriate prevention or treatment services. This supports the utility of the algorithm in enhancing targeted HIV testing efforts.

**Table 2. Clinical and demographic profile of patients, Zimbabwe, 2024. (N = 23,058).**

| Variable | Number | (%)* |
|---|---|---|
| **Total** | **23,058** | **(100)** |
| Age in years | | |
| ◦ 15–24 | 3,524 | (15.3) |
| ◦ 22–29 | 3,588 | (15.6) |
| ◦ 30–34 | 5,223 | (22.7) |
| ◦ 35–39 | 3,679 | (16.0) |
| ◦ 40–44 | 4,315 | (18.7) |
| ◦ 45–49 | 2,398 | (10.4) |
| ◦ >/= 50 | 331 | (1.4) |
| Gender | | |
| ◦ Male | 10,360 | (44.9) |
| ◦ Female | 12,698 | (55.1) |
| Risk Screened before testing | | |
| ◦ Yes | 15,289 | (66.3) |
| ◦ No | 7,769 | (33.7) |
| Type of HIV Test | | |
| ◦ First Test | 7,906 | (34.3) |
| ◦ Retest | 14,650 | (63.5) |
| ◦ Not Documented | 502 | (2.2) |
| HIV Test Result | | |
| ◦ Negative | 21,331 | (92.5) |
| ◦ Positive | 1,727 | (7.5) |
| Testing District | | |
| ◦ Mangwe | 1040 | (4.5) |
| ◦ Sanyati | 1232 | (5.3) |
| ◦ Gwanda | 1239 | (5.4) |
| ◦ Chipinge | 1350 | (5.9) |
| ◦ Mutare | 1254 | (5.4) |
| ◦ Gweru | 3575 | (15.5) |
| ◦ Kwekwe | 2443 | (10.6) |
| ◦ Chegutu | 3279 | (14.2) |
| ◦ Makonde | 2997 | (13.0) |
| ◦ Umzingwane | 1144 | (5.0) |
| ◦ Makoni | 2424 | (10.5) |
| ◦ Zvishavane | 1081 | (4.7) |

*Column percentage.

## Interpretation of key findings

This study offers significant insights into the effectiveness of status-neutral testing in targeted HIV testing in Zimbabwe.

Firstly, patients who underwent risk screening before HIV testing exhibited a higher positivity ratio of 8.5% compared to those who were not screened, who had a positivity ratio of 5.5%. The risk screening was conducted using a standardized algorithm, which is an integral part of the country's standard service delivery for determining eligibility for an HIV test. This

**Table 3. Factors associated with HIV Positivity among patients who tested for HIV, Zimbabwe, 2024. (N = 23,058).**

| Variable | Total | HIV Positive[#] | | OR (95 CI) | aOR (95 CI)[#] |
|---|---|---|---|---|---|
| | N | N | (%)[**] | | |
| **Total** | 23,058 | 1,727 | (7.5) | – | – |
| Age in years | | | | | |
| ◦ 15–24 | 3,524 | 440 | (12.5) | 7.43 (5.68, 9.73) | 7.42 (5.65, 9.86)^ |
| ◦ 25–29 | 3,588 | 72 | (2.0) | Ref | Ref |
| ◦ 30–34 | 5,223 | 92 | (1.8) | 1.69 (1.22, 2.33) | 1.68 (1.22, 2.34)^ |
| ◦ 35–39 | 3,679 | 334 | (9.1) | 10.32 (7.90, 13.49) | 10.31 (7.92, 13.54)^ |
| ◦ 40–44 | 4,315 | 342 | (7.9) | 12.26 (9.21, 16.31) | 12.23 (9.22, 16.36)^ |
| ◦ 45–49 | 2,398 | 410 | (17.1) | 23.61 (2.67, 4.89) | 3.61 (2.67, 4.90)^ |
| ◦ >/ = 50 | 331 | 29 | (8.8) | 5.07 (3.62, 7.11) | 5.06 (3.62, 7.12)^ |
| Gender | | | | | |
| ◦ Male | 10,360 | 912 | (8.8) | 3.10 (2.74, 3.49) | 3.09 (2.74, 3.49)^ |
| ◦ Female | 12,698 | 815 | (6.4) | Ref | Ref |
| Type of HIV Test | | | | | |
| ◦ Retest | 7,906 | 404 | (5.1) | Ref | Ref |
| ◦ First Test | 14,650 | 1,301 | (8.9) | 1.66 (1.44, 1.91) | 1.65 (1.43, 1.91)^ |
| ◦ Unspecified | 502 | 22 | (4.4) | 1.38 (0.91, 2.09) | |
| Linkage to Prevention[*] | | | | | |
| ◦ Screened | 15,289 | – | (66.3)[*] | 3.90 (3.05, 4.97) | 3.89 (3.05, 4.97)^ |
| ◦ Not Screened | 7,769 | – | (33.7) | Ref | Ref |

OR – Odds Ratio; aOR – adjusted Odds Ratio; CI – confidence interval.

[#]Positivity excludes missing variables.

[**]Row percentages; [#]Modified Poisson regression for aOR; ^p < 0.05, *** Fisher's Exact.

[*]Denominator is all tests done (N = 23,058), [##] Denominator is all Negative tests (N = 21,331).

finding underscores the importance of pre-test risk assessment in identifying individuals at higher risk of HIV infection and enhancing the efficiency of targeted testing strategies [17].

Furthermore, screened patients accounted for 75% of the overall positivity observed in the study, highlighting the effectiveness of targeted testing in Zimbabwe and underscoring the urgent need for accessible and effective HIV screening options, particularly for high-risk populations. This finding suggests that a significant proportion of individuals screened through traditional methods are indeed HIV-positive, indicating that many high-risk individuals are still not being reached effectively by current screening efforts. This is particularly relevant in the context of a generalized epidemic and declining funding for HIV programs, which has led to sporadic stockouts of HIV testing commodities. The results align with previous studies that emphasize the importance of using algorithms to assess risk and prioritize clients for HIV testing while offering self-screening options to those at lower risk [17,23–25]. To remain on course of achieving and sustaining the targets for case identification, it is therefore imperative to effectively implement an algorithm that aids health workers in prioritizing patients who are most likely to test HIV positive

Secondly, 98.7% (n = 13,811) of the clients who tested HIV negative following risk screening were linked to HIV prevention services. In multivariate analysis, the probability of linkage among HIV-negative screened clients was 3.89 (adjusted 95% CI: 3.05, 4.97). This finding indicates that screening clients for testing facilitates focused HIV prevention linkage among those who test negative. Our findings are consistent with the status-neutral approach, which emphasizes equal importance on linkage to both prevention and treatment services.

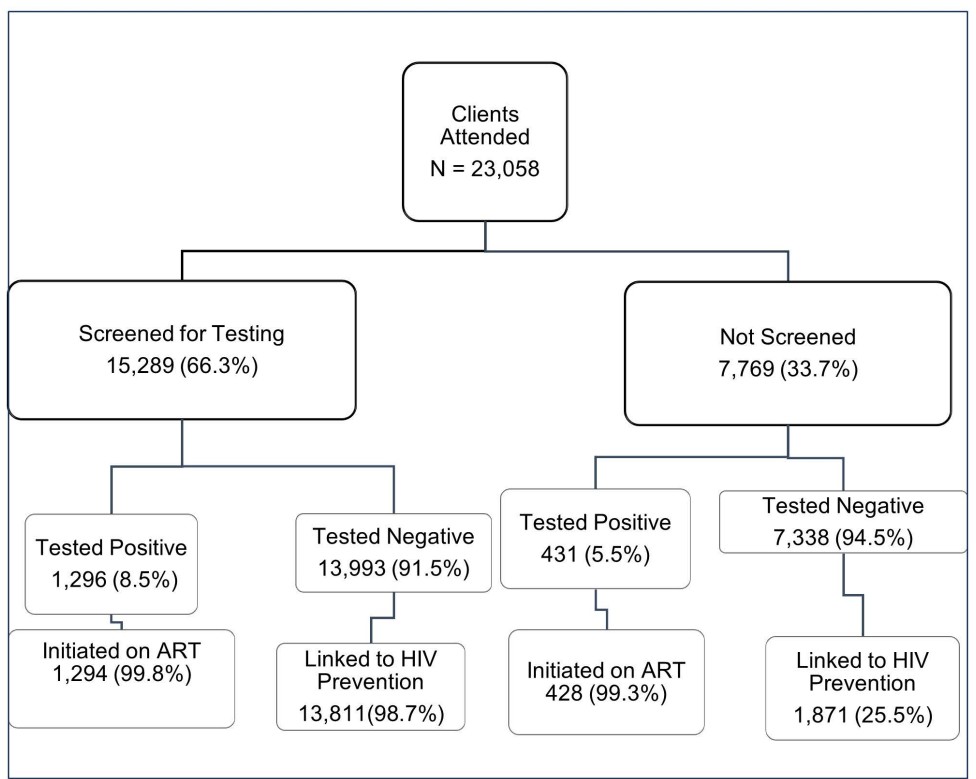

**Fig 2. HIV Testing and Post-test Linkages, Zimbabwe, 2024.**

[6,7,26,27]. However, our approach differs from other studies that place risk assessment after testing rather than before. Conducting risk screening post-test increases the number of clients in the post-test stage with an unknown risk profile, necessitating additional risk screening. This increased workload, particularly in high-volume and high-frequency testing contexts, may compromise the thoroughness of the screening process [28,29].

This study highlights the utility of conducting risk screening before HIV testing within the status-neutral framework. This approach effectively complements the targeted testing strategy by reducing the frequency of testing that does not align with the individual's risk profile. By embedding the status-neutral concept into targeted testing, we ensure that testing efforts are focused on those at higher risk, thereby optimizing resource use and improving the efficiency of HIV testing programs.

Implementing the status-neutral concept without integrating it into targeted testing deviates from the standardized retesting algorithm, which recommends a maximum testing frequency of once every three months for individuals at ongoing risk of HIV infection. This integration is crucial for maintaining adherence to established guidelines while enhancing the effectiveness of HIV prevention and care strategies [7,30].

Third, 63.5% of the clients who tested for HIV were retests (n = 14,650), and yet first tests were 65% more likely to test HIV positive (a95%CI: 1.43, 1.91), adjusted for age and sex. Most tests being retested may be suggestive of "over-testing" or high-risk perception, particularly given the low positivity ratios obtained [4]. Clients who test HIV negative at contact are retested annually if they fall into the general population category and retested 3 monthly if they are at ongoing risk for HIV transmissions, such as sero-different couples and those on PrEP, according to the national retesting algorithm [31].

Finally, our adjusted analysis revealed that men were 3.09 times more likely to test HIV positive (adjusted 95% CI: 2.74, 3.49), despite representing only 44.9% (10,360/23,058) of the tested population. This finding suggests that a smaller proportion of men undergo HIV testing, yet a higher percentage of those who do test positive. This observation is consistent with previous studies and widely available public information [32,33]. Additionally, men are less likely to adhere to HIV treatment and are more prone to unfavourable treatment outcomes, which can be attributed to various factors, including prevailing gender norms [31,34].

## Implications for policy and practice

Targeted testing remains the cornerstone of HIV Testing Services (HTS) programming aimed at achieving epidemic control. When complemented with a status-neutral approach to HIV testing, it creates a dual strategy that prioritizes both testing and linkage to prevention and treatment services. This combined approach ensures that individuals who test positive are promptly linked to care, while those who test negative are connected to prevention services, thereby reducing the risk of new infections. Vigilant implementation of this concept is crucial to expedite epidemic control and enhance the overall effectiveness of HIV prevention and treatment efforts.

To effectively implement the status-neutral targeted HIV testing approach in Zimbabwe and beyond, several actionable steps and policy changes are necessary. Key stakeholders must also be identified to drive this initiative forward. Below are the recommended steps, policy adjustments, and potential barriers:

*Actionable Steps*:

 i. *Policy Development and Alignment*: Develop national policies that support the integration of the status-neutral approach with targeted HIV testing. These policies should align with existing health strategies and frameworks to ensure seamless implementation.

 ii. *Stakeholder Engagement*: Engage key stakeholders, including government health departments, non-governmental organizations (NGOs), community leaders, healthcare providers, and international partners such as the World Health Organization (WHO) and UNAIDS. Collaborative efforts are essential for resource mobilization, advocacy, and sustained support.

 iii. *Capacity Building*: Invest in the training and capacity building of healthcare providers to enhance their understanding and implementation of the integrated approach. Training programs should focus on the benefits, procedures, and best practices for combining status-neutral and targeted testing methods.

 iv. *Resource Allocation*: Ensure adequate resources are available for the implementation of the dual strategy. This includes funding for testing kits, infrastructure, personnel, and support services necessary for continuous engagement in care.

 v. *Community Outreach and Education:* Launch community outreach programs to raise awareness about the integrated approach. Educational campaigns should aim to reduce stigma, encourage testing, and promote the benefits of continuous care and targeted interventions.

Policy Adjustments:

 i. *Regulatory Support*: Implement regulatory frameworks that facilitate the integration of the status-neutral approach into existing HIV testing guidelines. This may involve revising existing protocols and ensuring compliance with new regulations.

ii. *Data Management and Monitoring*: Establish robust data management systems to monitor the implementation and effectiveness of the integrated approach. Regular monitoring and evaluation will help identify gaps, measure progress, and inform policy adjustments.

Potential Barriers:

i. *Limited Resources*: Insufficient funding and resources can impede the effective rollout of the integrated approach. Securing sustainable funding and resource allocation is critical.

ii. *Healthcare Infrastructure*: Inadequate healthcare infrastructure, particularly in rural areas, can challenge the delivery of services. Investments in infrastructure improvements are necessary.

iii. *Data Privacy Concerns*: Ensuring the confidentiality and security of patient data is paramount. Robust data protection measures must be implemented to build trust and encourage participation.

By addressing these steps, policies, stakeholders, and barriers, the integrated status-neutral targeted HIV testing approach can be effectively implemented, leading to improved health outcomes and enhanced HIV prevention and treatment efforts in Zimbabwe and beyond.

## Strengths

The availability of primary source documents, such as HTS registers at all visited facilities facilitated the data abstraction process. In addition, the sampled 36 health facilities provided a large sample size that enabled us to draw inferences on the population of the country.

## Limitations

Discrepancies between data abstracted from HTS registers, monthly summaries and DHIS2 during data triangulation exposed data entry or computation errors that could be rectified by onsite data analysis, and cascade generation.

## Conclusions

Our study demonstrates that targeted testing and status-neutral testing are complementary concepts that, when applied together, can enhance the identification and management of people living with HIV. By prioritizing high-risk individuals for testing and ensuring effective linkage to both prevention and treatment services, this dual approach can help meet case identification targets and reduce ongoing HIV transmission. This strategy not only improves health outcomes but also ensures the efficient use of limited resources, particularly in low- and middle-income countries.

*Recommendations for Future Research* Future research should focus on the long-term effectiveness, cost-efficiency, and integration of this approach with other health services, as well as the role of technology and addressing barriers to implementation. These steps aim to improve HIV prevention and treatment outcomes, contributing to better healthcare delivery and public health.

## Supporting information

**S1 File. Status neutral paper dataset.**
(XLSX)

## Acknowledgments

I acknowledge several individuals and institutions that made this study a success. Special gratitude goes to my academic supervisors, Professor M. Tshimanga, Dr J. Chirenda and Dr K. Takarinda, The Director of AIDS & TB Unit, Dr O. Mugurungi and the entire HTS team for their support and prodding during this study.

## Author contributions

**Conceptualization:** Hamufare D. Mugauri, Owen Mugurungi, Joconiah Chirenda, Kudakwashe Takarinda, Prosper Mangwiro, Mufuta Tshimanga.

**Data curation:** Hamufare D. Mugauri, Owen Mugurungi, Joconiah Chirenda, Kudakwashe Takarinda, Mufuta Tshimanga.

**Formal analysis:** Hamufare D. Mugauri, Owen Mugurungi, Joconiah Chirenda, Prosper Mangwiro, Mufuta Tshimanga.

**Funding acquisition:** Hamufare D. Mugauri, Mufuta Tshimanga.

**Investigation:** Hamufare D. Mugauri, Kudakwashe Takarinda.

**Methodology:** Hamufare D. Mugauri, Joconiah Chirenda, Kudakwashe Takarinda.

**Project administration:** Hamufare D. Mugauri.

**Resources:** Hamufare D. Mugauri.

**Software:** Hamufare D. Mugauri, Kudakwashe Takarinda.

**Supervision:** Hamufare D. Mugauri, Joconiah Chirenda, Kudakwashe Takarinda, Mufuta Tshimanga.

**Validation:** Hamufare D. Mugauri, Kudakwashe Takarinda, Prosper Mangwiro, Mufuta Tshimanga.

**Visualization:** Hamufare D. Mugauri, Owen Mugurungi, Kudakwashe Takarinda, Prosper Mangwiro, Mufuta Tshimanga.

**Writing – original draft:** Hamufare D. Mugauri, Prosper Mangwiro.

**Writing – review & editing:** Hamufare D. Mugauri, Owen Mugurungi, Joconiah Chirenda, Kudakwashe Takarinda, Prosper Mangwiro, Mufuta Tshimanga.

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
