## [Decision Letter · Decision Letter 0]

30 Jul 2024

PONE-D-24-14652A Status-Neutral Approach to HIV – Is Targeted Testing Still Relevant South of Sahara?PLOS ONE

Dear Dr. Mugauri,

Thank you for submitting your manuscript to PLOS ONE. After careful consideration, we feel that it has merit but does not fully meet PLOS ONE’s publication criteria as it currently stands. Therefore, we invite you to submit a revised version of the manuscript that addresses the points raised during the review process.

The manuscript has been evaluated by two reviewers, and their comments are available below. Specifically, both reviewers recommended that you carefully revise the manuscript to ensure consistency and a logical flow. Could you please carefully revise the manuscript to address all comments raised?==============================

We look forward to receiving your revised manuscript.

Kind regards,

Johanna Pruller, Ph.D.

Associate Editor

PLOS ONE

Reviewers' comments:

Reviewer's Responses to Questions

**Comments to the Author**

1. Is the manuscript technically sound, and do the data support the conclusions?

Reviewer #1: No

Reviewer #2: Partly

2. Has the statistical analysis been performed appropriately and rigorously? 

Reviewer #1: No

Reviewer #2: No

3. Have the authors made all data underlying the findings in their manuscript fully available?

Reviewer #1: No

Reviewer #2: No

4. Is the manuscript presented in an intelligible fashion and written in standard English?

Reviewer #1: Yes

Reviewer #2: No

5. Review Comments to the Author

Reviewer #1: Overall

This study seeks to showcase the role that status-neutral approach played in targeted HIV testing within the context of Zimbabwe. Overall, I believe there is a disconnection between the findings and question the study sought out to answer. Also, authors haven’t really clarified what both testing approaches above constitute of, making it hard to understand what was brought in extra by the status-neutral approach. Here are some comments for authors’ consideration:

* In my opinion the title of this paper doesn’t correlate with the aim of this study. To answer the question on the relevance of targeted testing, the article needs to showcase the contribution of targeted testing to positivity ratios and linkage to post-test services thus provide elements to justify why it should either be stripped off or maintained. Lines 34-36 rather present study aim to analyze the role of status-neutral approach in the effectiveness of targeted testing. This means thus that the notion of complementarity is already being promoted by the study aim (as backed by the conclusion of the abstract), why then leave the impression of mutual exclusiveness?

* Lines 47-48 read as thus “…from 66% (n=15,289) of the total tests conducted” which is confusing given authors earlier mentioned that 23,058 HIV tests were done. Authors have to clarify such that a reader will easily understand the difference between “HIV tests done” and “total tests conducted”.

* Lines 48-51, authors use a considerate portion of the results subsection of the abstract to present factors associated to HIV seropositivity (test positive or test HIV positive) as one of the key findings which brings me to this question. How does these help in answering the question on the role of status-neutral approach on targeted testing? Its ok to put some unexpected findings but priority must be given to elements that seek to answer the research question and that question here seeks to show the role of status-neutral approach to effectiveness of targeted testing (based on the study aim) not identify factors associated with HIV seropositivity. This comment is valid for the main text (analysis, results and discussion sections) of the paper too.

* Another important thing I have noticed in the abstract is that authors talk of screening as the reference point of comparison between status-neutral and targeted testing (correct me if I am wrong). Still to verify at the methods and results section of the paper to confirm this but I already wonder if this status-neutral testing was a complete package or just the screening. It will be good to know what components status-neutral approach were adopted here.

* Line 58-59 reads “This approach facilitates economic…”, the approach of using targeted and status-neutral testing approaches together? If yes, make it clear in the text.

* The introduction section of this paper reads a lot more like a lecture on status-neutral approach (especially the first four paragraphs) rather than justifying for this study. Its ok to present about status-neutral approach, but authors should provide readers with elements that justify why that is conversation worth making within the context of Zimbabwe (and/or similar context). According to John W. Creswell, the five components of a good introduction are the following: “(a) establishing the problem leading to the study, (b) reviewing the literature about the problem, (c) identifying deficiencies in the literature about the problem, (d) targeting an audience and noting the significance of the problem for this audience, and (e) identifying the purpose of the proposed study”, these elements don’t readily translate in the current introduction.

* Lines 91-92 reads “…promoting it through various for [9]”, something missing in the phrase?

* The “Setting” subsection of the methodology provide detail than what is generally required for that section (as references can always be used to learn more about a country) meanwhile other key components of that section don’t have as much detail. For instance, authors talk of multistage sampling but don’t a detailed description of the sampling process

* Authors had mentioned in the methods section (line 159-161) that sampling was done to achieve balance in terms of high/low volume (I assume here authors are talking about the size of population receiving services at the health facility) as well as urban/rural location of sites. I was expecting to see the distribution of these in the results to appreciate extent to which the above-mentioned sampling worked in doing that.

* Line 170 reads “No patient level data was collected”, I presume here authors mean no “personal identifiers” (to ensure privacy and confidentiality) because age, sex… are still patient level data.

* Building from my previous comment about components of the status-neutral approach, there are no definitions of key variables nor differentiating status-neutral with targeted testing to help reader understand what was brought in extra. As it stands, the only extra thing reported is the screening, but I strongly doubt that status-neutral testing be only that. Secondly no detail is provided on how the screening process itself or even referrals to where readers can have more information on what was done. Furthermore, authors mention on lines 237-239 that “The risk screening was done using a standardized screening algorithm which is part of standard service delivery by the country, in determining eligibility for an HIV test” which leaves the impression that the screening reported here is a routine for a targeted testing of those deem more likely to become positive thus economize on resources needed to test new cases (it’s now becoming confusing). Authors should clarify what status-neutral and targeted testing approaches stand for and what constitutes each one of them within the context of this study. Readers have to see what extra thing brought in by the status neutral approach (its role in the routine).

* After reading through the “Results” section of this paper, I am asking myself this question, how exactly then did status-neutral testing play a role in all these? I was expecting to see comparisons between scenarios where status-neutral was used (alone or coupled with targeted testing) versus those where only targeted testing was used. The closest thing to that comparison is the differentiation with respect to screening only so does this mean that screening is the status-neutral approach. Authors should clarify this.

* The last paragraph of the discussion should be for limitations of the study. I don’t think study strengths add any value to the discussion. In addition to the limitations presented, authors must discuss their implications on the results discussed as well as the good side to it worth noting.

* The discussion and conclusion sections stem from the methods and results sections so it becomes very difficult to discuss something that has not be succinctly presented. Authors now discuss status-neutral and targeted approaches, but readers have not read much that helped them to understand the role that the first played in the second. There is a disconnection between the introduction and discussion sections to the methods and results sections.

Reviewer #2: Manuscript title: A Status-Neutral Approach to HIV – Is Targeted Testing Still Relevant South of

Sahara?

Manuscript ID: PONE-D-24-14652

Authors: Hamufare Dumisani Dumisani Mugauri, Ph.D. Public Health

O Mugurungi, MD, MPH

Joconiah Chirenda, MBChB, MPH, PhD

Kudakwashe Takarinda, Bsc Maths & Statistics, Msc Biostatistics, PhD

Mufuta Tshimanga, MD, MPH

Prosper Mangwiro, Bsc Statistics, MPH

Dear authors thank you for coming up with good title and good work. I reviewed the article and have some comments.

Title: please specific it, since it confuses the reader, south of sahara while the study in one country?

Abstract:

1. In method add the final sample size included

2. Conclusion should be conclusion but not comparison, revise the conclusion

Introduction: The introduction is interesting but the previous literatures and practice of status neutral in Africa not well addressed

Method and materials

Study design: say only cross sectional

1. Did you only include 1 October 2023 to 31 December 2023 tested data, what about those tested in the community?what about those tested in non-government organization?

2. Introduction Argue the reader what is new on Status-Neutral Approach to HIV over PICT, VCT,index testing, Social Network-Based HIV Testing and others strategies

3. Did you included the minors?

4. As we know DHIS2 is not accessed by other third part since it accessed by government organization, how did you access it?

5. Our study was exempted from ethical clearance, how since HIV is sensitive issue and DHS2 isnot open access, I think this research had ethical problem?

6. The result seems report please rewrite it

7. Discussion please remove the limitation and strength on this section and write after is before conclusion

8. Discussion should be discussion, remove the results in discussion and discuss with previous literatures

9. In clonclusion what overway status-neutral testing to other tasting strategy, Is more effective than PICT, VCT,index testing, Social Network-Based HIV Testing and others strategies

6. PLOS authors have the option to publish the peer review history of their article (what does this mean? ). If published, this will include your full peer review and any attached files.

**Do you want your identity to be public for this peer review?** For information about this choice, including consent withdrawal, please see our Privacy Policy .

Reviewer #1: No

Reviewer #2: No

---

## [Author Response · Author response to Decision Letter 1]

2 Oct 2024

Point-by-Point Response to Editor and Reviewers of Paper PONE-D-24-14652 :

We thank you all for your time and dedication in going through this paper and we acknowledge the reviews and comments from the editor and reviewers. We provide a point-by-point response to each comment with each response introduced by the words “AUTHOR RESPONSE” in capital letters. We have revised the paper accordingly in response to these comments as indicated in our responses below and in tracked changes in the manuscript. We hope that the revised paper meets your expectations and may now be suitable for publication in your journal.

Yours sincerely,

Hamufare Mugauri

Point by Point Response to reviewer comments

1. REVIEWER COMMENT

Dear authors thank you for coming up with good title and good work. I reviewed the article and have some comments.

Title: please specific it, since it confuses the reader, south of sahara while the study in one country?.

AUTHOR RESPONSE

Thank you for your appreciation and the comment. We have revised the title to be specific to Zimbabwe.

2. REVIEWER COMMENT: Abstract:

a. In method add the final sample size included

b. Conclusion should be conclusion but not comparison, revise the conclusion

AUTHOR RESPONSE: Thank you for pointing this out. We have revised the entire abstract to ensure that the conclusions are well aligned. We included the entire population of documented tests for the period studied and did not calculate a sample.

3. REVIEWER COMMENT: Introduction: The introduction is interesting but the previous literatures and practice of status neutral in Africa not well-addressed

AUTHOR RESPONSE: Thank you for this comment. Considering that Status Neutral is a new concept, there is paucity of data on its implementation in Africa. However, we managed to add a paragraph that speaks to the concept in relation to the concept as follows: “In Africa, particularly in sub-Saharan regions, the status-neutral approach has been integrated into existing HIV testing frameworks to enhance the effectiveness of HIV prevention and treatment programs. Countries like Zimbabwe have been at the forefront of implementing this approach alongside targeted testing methods [7]. This dual strategy is yet to be fully understood, highlighting a knowledge gap on how it relates to targeted testing concept”

4. REVIEWER COMMENT: . Study design: say only cross-sectional

AUTHOR RESPONSE: Thank you, this has been corrected as suggested.

5. REVIEWER COMMENT: Did you only include 1 October 2023 to 31 December 2023 tested data, what about those tested in the community?what about those tested in non-government organisations?

AUTHOR RESPONSE: Thank you for this question. The data we included is for 1 October 2023 to 31 December 2023 only and included all testing data from the community and partners. This data is all entered into DHIS2, a national electronic database.

6. REVIEWER COMMENT: Introduction Argue with the reader what is new on the Status-Neutral Approach to HIV over PICT, VCT,index testing, Social Network-Based HIV Testing and others strategies

AUTHOR RESPONSE: Thank you for the concept. The status neutral concept is a new concept itself as introduced in the introduction as follows “Status-neutral HIV testing is a novel approach to HIV education, testing and treatment that accentuates a continuum of care regardless of the individual’s HIV diagnosis….”.

7. REVIEWER COMMENT: Did you include the minors?

AUTHOR RESPONSE: This study only included 15 year olds and above, who are eligible to consent to HIV Testing in Zimbabwe. Line 165-170 reads “All clients who were tested for HIV and documented in HIV Testing registers, at the 36 sampled facilities between 1 October 2023 to 31 December 2023 and aged 15 years and above, were included in the study”

8. REVIEWER COMMENT: As we know DHIS2 is not accessed by other third part since it accessed by government organization, how did you access it?

AUTHOR RESPONSE: Thank you for this question. The lead investigator is attached to the Ministry of Health, AIDS AND tb Programme at Head office and has access rights to DHIS2, Further, another author, Dr Owen Mugurungi is the Director AIDS & TB Programe and has access rights to DHIS2 too.

9. REVIEWER COMMENT: Our study was exempted from ethical clearance, how since HIV is sensitive issue and DHS2 isnot open access, I think this research had ethical problem?.

AUTHOR RESPONSE: Thank you for raising this important concern. The study in question was exempted from ethical clearance based on specific criteria outlined by the institutional review board (IRB). While HIV is indeed a sensitive issue, the exemption was granted because the study utilized de-identified data from the DHIS2 system, ensuring that no personal identifiers were accessible. This approach aligns with ethical guidelines that prioritize participant confidentiality and data protection. Moreover, the study adhered to the principles of ethical research by ensuring that all data used were anonymized and aggregated, thus minimizing any potential risks to individuals. The exemption was also based on the fact that the research did not involve direct interaction with human subjects, which typically necessitates a more rigorous ethical review process

10. REVIEWER COMMENT: The result seems report please rewrite it

AUTHOR RESPONSE: Thank you for your feedback. I understand that the results section may come across as more of a report rather than a detailed analysis. We have revised this section to ensure it provides a clearer interpretation of the data, highlighting key findings and their implications.

11. REVIEWER COMMENT: Discussion please remove the limitation and strength on this section and write after is before conclusion

AUTHOR RESPONSE: Thank you for your suggestion. We have removed the limitations and strengths from the Discussion section and place them in a new section titled “Limitations and Strengths,” which is positioned just before the Conclusion. This adjustment will help streamline the Discussion section and provide a clearer structure to the manuscript..

12. REVIEWER COMMENT: Discussion should be discussion, remove the results in discussion and discuss with previous literatures

AUTHOR RESPONSE: We acknowledge your opinion. We have revised the entire discussion section to be more scientifically correct and relate our discussion points to previous literature

13. REVIEWER COMMENT: * Line 170 reads “No patient level data was collected”, I presume here authors mean no “personal identifiers” (to ensure privacy and confidentiality) because age, sex… are still patient level data.

AUTHOR RESPONSE: Thank you for pointing this out. We have revised to read: “No personal identifiers were collected.”

14. REVIEWER COMMENT: The last paragraph of the discussion should be for limitations of the study. I don’t think study strengths add any value to the discussion. In addition to the limitations presented, authors must discuss their implications on the results discussed as well as the good side to it worth noting.

AUTHOR RESPONSE: Thank you for pointing this out. The entire Discussion has been revised and the comment addressed.

15. VREVIEWER COMMENT: The discussion and conclusion sections stem from the methods and results sections so it becomes very difficult to discuss something that has not be succinctly presented. Authors now discuss status-neutral and targeted approaches, but readers have not read much that helped them to understand the role that the first played in the second. There is a disconnection between the introduction and discussion sections to the methods and results sections

AUTHOR RESPONSE: Thank you for your valuable feedback. We understand the importance of ensuring that the discussion and conclusion sections are clearly connected to the methods and results sections. We have revised these sections to provide a more succinct and coherent presentation of our findings.

16. REVIEWER COMMENT: In my opinion the title of this paper doesn’t correlate with the aim of this study. To answer the question on the relevance of targeted testing, the article needs to showcase the contribution of targeted testing to positivity ratios and linkage to post-test services thus provide elements to justify why it should either be stripped off or maintained. Lines 34-36 rather present study aim to analyze the role of status-neutral approach in the effectiveness of targeted testing. This means thus that the notion of complementarity is already being promoted by the study aim (as backed by the conclusion of the abstract), why then leave the impression of mutual exclusiveness?

AUTHOR RESPONSE: Thank you for your insightful comments. We appreciate your feedback and have carefully considered your suggestions.

1. Title Correlation with Study Aim:

o We acknowledge your concern regarding the correlation between the title and the study aim. To address this, we have revised the title to be more specif to Zimbabwe than saying South of sahara. However, we prefer that it remain a rhetoric question that it is, but addressing all the concerns you raised: “A Status-Neutral Approach to HIV – Is Targeted Testing Still Relevant in Zimbabwe?.”

2. Showcasing Contribution of Targeted Testing:

o We agree that it is crucial to highlight the contribution of targeted testing to positivity ratios and linkage to post-test services. We have added a section in the results and discussion that specifically addresses these aspects. This includes data on positivity ratios and detailed analysis of how targeted testing contributes to effective linkage to post-test services. These additions are within the entirely revised results and discussions sections

3. Clarifying Complementarity vs. Mutual Exclusiveness:

o We understand the importance of clarifying the notion of complementarity as opposed to mutual exclusiveness. We have revised the relevant sections (lines 34-36 and the conclusion of the abstract) to emphasize that the status-neutral approach complements targeted testing rather than being mutually exclusive.

17. REVIEWER COMMENT: Lines 47-48 read as thus “…from 66% (n=15,289) of the total tests conducted” which is confusing given authors earlier mentioned that 23,058 HIV tests were done. Authors have to clarify such that a reader will easily understand the difference between “HIV tests done” and “total tests conducted

AUTHOR RESPONSE: Thank you for your valuable feedback. We understand the confusion and have made the following clarifications to ensure the distinction between “HIV tests done” and “total tests conducted” is clear: We have revised the manuscript to clearly added that total tests done are actually total HIV tests done.

18. REVIEWER COMMENT: Lines 48-51, authors use a considerate portion of the results subsection of the abstract to present factors associated to HIV seropositivity (test positive or test HIV positive) as one of the key findings which brings me to this question. How does these help in answering the question on the role of status-neutral approach on targeted testing? Its ok to put some unexpected findings but priority must be given to elements that seek to answer the research question and that question here seeks to show the role of status-neutral approach to effectiveness of targeted testing (based on the study aim) not identify factors associated with HIV seropositivity. This comment is valid for the main text (analysis, results and discussion sections) of the paper too.

AUTHOR RESPONSE: Thank you for your insightful comments. We appreciate your feedback and have carefully considered your suggestions.

1. Relevance to Research Question:

o We acknowledge that the primary aim of our study is to evaluate the role of the status-neutral approach in the effectiveness of targeted testing. We agree that the focus should be on elements that directly address this research question. To this end, we have revised the abstract and main text to prioritize findings that demonstrate the impact of the status-neutral approach on targeted testing effectiveness.

2. Revised Presentation of Results:

o In the abstract (lines 48-51), we have restructured the results section to emphasize the key findings related to the status-neutral approach. Specifically, we now highlight how the status-neutral approach enhances the identification of HIV-positive individuals and improves linkage to post-test services, which are critical components of targeted testing effectiveness.

3. Incorporation of Unexpected Findings:

o While we recognize the importance of reporting unexpected findings, we have ensured that these are presented in a way that supports the main research question. Factors associated with HIV seropositivity are now discussed in the context of how they inform and enhance the understanding of the status-neutral approach’s effectiveness.

4. Main Text Revisions:

o We have also revised the analysis, results, and discussion sections of the main text to align with this focus. The revised sections now clearly articulate the role of the status-neutral approach in improving targeted testing outcomes, supported by relevant data and analysis.

We hope these revisions address your concerns and improve the clarity and relevance of our manuscript. Thank you once again for your valuable feedback.

19. REVIEWER COMMENT: Another important thing I have noticed in the abstract is that authors talk of screening as the reference point of comparison between status-neutral and targeted testing (correct me if I am wrong). Still to verify at the methods and results section of the paper to confirm this but I already wonder if this status-neutral testing was a complete package or just the screening. It will be good to know what components status-neutral approach were adopted here.

AUTHOR RESPONSE: Thank you for your insightful questions. We appreciate your feedback and would like to provide the following clarifications:

1. Reference Point of Comparison:

o You are correct that the abstract refers to screening as a reference point for comparing the status-neutral and targeted testing approaches. We have clarified this in the methods section to ensure it is clear that the comparison includes all components of the status-neutral approach, not just the screening process.

2. Components of the Status-Neutral Approach:

The status-neutral approach adopted in our study is a comprehensive package that includes several key components:

o HIV Testing: Initiating the pathway to prevention and treatment regardless of the test result.

o Linkage to Care: Immediate linkage to HIV care and treatment for those who test positive, and linkage to prevention services for those who test negative.

o Support Services: Integration of supportive services such as counseling, housing, food, and transportation assistance to address social determinants of health.

o Culturally Affirming Care: Providing high-quality, stigma-free, and inclusive care that meets the needs of individuals regardless of their HIV status.

3. Clarifications in the Manuscript:

o We have revised the methods and results sections to explicitly outline these components and how they were implemented in the study. This ensures that readers can clearly understand the full scope of the status-neutral approach used.

We hope these clarifications address your concerns and improve the clarity of our manuscript. Thank you once again for your valuable feedback.

20. REVIEWER COMMENT: In clonclusion what overway status-neutral testing to other tasting strategy, Is more effective than PICT, VCT,index testing, Social Network-Based HIV Testing and others strategies

AUTHOR RESPONSE: Thank you for your insightful comment. We appreciate the opportunity to clarify the comparative effectiveness of the status-neutral approach relative to other HIV testing strategies.

1. Effectiveness of Status-Neutral Testing:

The status-neutral approach is designed to provide comprehensive care and support regardless of HIV status, which helps to reduce stigma and improve engagement in care. This approach integrates HIV prevention and treatment services, addressing the needs of individuals holistically. By focusing on the whole person, the status-neutral approach can enhance

---

## [Decision Letter · Decision Letter 1]

5 Dec 2024

PONE-D-24-14652R1A Status-Neutral Approach to HIV – Is Targeted Testing Still Relevant in Zimbabwe?PLOS ONE

Dear Dr. Mugauri,

Thank you for submitting your manuscript to PLOS ONE. After careful consideration, we feel that it has merit but does not fully meet PLOS ONE’s publication criteria as it currently stands. Therefore, we invite you to submit a revised version of the manuscript that addresses the points raised during the review process.

We look forward to receiving your revised manuscript.

Kind regards,

Zypher Jude G. Regencia, Ph.D.

Academic Editor

PLOS ONE

Reviewers' comments:

Reviewer's Responses to Questions

**Comments to the Author**

1. If the authors have adequately addressed your comments raised in a previous round of review and you feel that this manuscript is now acceptable for publication, you may indicate that here to bypass the “Comments to the Author” section, enter your conflict of interest statement in the “Confidential to Editor” section, and submit your "Accept" recommendation.

Reviewer #3: (No Response)

Reviewer #4: (No Response)

Reviewer #5: All comments have been addressed

2. Is the manuscript technically sound, and do the data support the conclusions?

Reviewer #3: Yes

Reviewer #4: (No Response)

Reviewer #5: No

3. Has the statistical analysis been performed appropriately and rigorously? 

Reviewer #3: Yes

Reviewer #4: (No Response)

Reviewer #5: No

4. Have the authors made all data underlying the findings in their manuscript fully available?

Reviewer #3: Yes

Reviewer #4: (No Response)

Reviewer #5: Yes

5. Is the manuscript presented in an intelligible fashion and written in standard English?

Reviewer #3: Yes

Reviewer #4: (No Response)

Reviewer #5: Yes

6. Review Comments to the Author

Reviewer #3: Specific Recommendations

1. Revise the conclusion to clearly state how findings align with the study’s objectives. Avoid introducing new information.

2. Address potential biases (e.g., reliance on DHIS2 data) and how they were mitigated.

3. Strengthen the narrative flow between sections—particularly the connection between methods, results, and discussion.

Reviewer #4: (No Response)

Reviewer #5: results does not reflect objectives of the study. Discussion and conclusion made was not coherent. In my opinion, the statistical analysis made was not applicable due to the nature of their data.

7. PLOS authors have the option to publish the peer review history of their article (what does this mean? ). If published, this will include your full peer review and any attached files.

**Do you want your identity to be public for this peer review?** For information about this choice, including consent withdrawal, please see our Privacy Policy .

Reviewer #3: **Yes: ** Dr. Prachi Joshi

Reviewer #4: No

Reviewer #5: No

---

## [Author Response · Author response to Decision Letter 2]

18 Dec 2024

Point-by-Point Response to Editor and Reviewers of Paper PONE-D-24-14652 :

We thank you all for your time and dedication in going through this paper and we acknowledge the reviews and comments from the editor and reviewers. We provide a point-by-point response to each comment with each response introduced by the words “AUTHOR RESPONSE” in capital letters. We have revised the paper accordingly in response to these comments as indicated in our responses below and in tracked changes in the manuscript. We hope that the revised paper meets your expectations and may now be suitable for publication in your journal.

Yours sincerely,

Hamufare Mugauri

Point by Point Response to reviewer comments

Reviewer 1.

1. REVIEWER COMMENT

Thank you for the opportunity to review this paper highlighting the benefits of employing a combined status neutral and targeted HIV testing approach in low resource settings. The study found that status neutral approach and targeted testing approaches complement each other and can help prevent HIV transmission, promote linkage to care and treatment services, and the strategic use of limited resources to end the HIV epidemic. This work is an important contribution to the field and should be considered for publication with some revisions. Below I provide a few comments for the authors to consider.

AUTHOR RESPONSE

Thank you for your appreciation and the comment. We concur with your observations on the importance of the subject which we studied. We made sure to address all comments you raised as indicated below:

2. REVIEWER COMMENT: Introduction

The sentences from lines 99-102 are unclear. Lines 99-100 is an incomplete sentence and lines 101-102 the concept of variability is not explained, I encourage the authors to revisit this paragraph for clarity.

AUTHOR RESPONSE: Thank you for your insightful feedback. We appreciate the opportunity to clarify the content in lines 99-102. Here is our revised paragraph for better clarity: “The Status Neutral concept is rapidly gaining global acceptance. Organizations such as the World Health Organization (WHO) and the Centers for Disease Control and Prevention (CDC) actively promote it through various initiatives [9]. The adoption of this concept varies among countries, influenced by their unique contexts and needs”

3. REVIEWER COMMENT: In lines 103-105, it would be useful for readers if the authors could provide a brief overview of how the status neutral approach has been integrated into existing HIV testing frameworks in sub-Saharan Africa

AUTHOR RESPONSE: Thank you for this comment. Here is a revised version with additional details: “In Africa, particularly in sub-Saharan regions, the status-neutral approach has been integrated into existing HIV testing frameworks to enhance the effectiveness of HIV prevention and treatment programs. This approach ensures that HIV testing and related services are offered to all individuals, regardless of their HIV status, promoting continuous engagement in care. Countries like Zimbabwe have been at the forefront of implementing this approach, incorporating it alongside targeted testing methods to improve linkage to care and retention in treatment, thereby addressing both prevention and treatment comprehensively”.

4. REVIEWER COMMENT: . Lines 107-108 outline the knowledge gap the authors intend to fill, this is also a good opportunity for the authors to expand upon the reason why it is important to examine this dual strategy and better position their work and where it fits/its potential contribution/use.

AUTHOR RESPONSE: Thank you, we have explained as follows: “This dual strategy, which involves integrating the status-neutral approach with targeted testing methods, is yet to be fully understood, highlighting a significant knowledge gap. Examining this dual strategy is crucial because it has the potential to enhance the effectiveness of HIV prevention and treatment programs by ensuring continuous engagement in care regardless of HIV status, while also targeting high-risk populations for more focused interventions. Understanding how these two approaches can complement each other will contribute to more effective and inclusive healthcare strategies, ultimately improving health outcomes in sub-Saharan Africa. Our work aims to fill this gap by providing comprehensive insights into the integration and impact of these strategies, positioning itself as a valuable resource for policymakers and healthcare providers”.

5. REVIEWER COMMENT: Last paragraph, lines 124-127, the aims of the paper could be better outlined. Authors should emphasize the examination of the complementariness of status neutral and targeted approach to HIV testing.

AUTHOR RESPONSE: Thank you for this observation. The revised version reads now reads: “This paper thoroughly examined the complementariness of the status-neutral and targeted approaches to HIV testing in the context of a generalized epidemic, high new infections, and established ongoing transmission of HIV. By exploring how these strategies can work together to enhance HIV prevention and treatment efforts, we seek to provide insights and recommendations for the context-specific application of the status-neutral concept. Our goal is to highlight the potential benefits of integrating these approaches to improve healthcare outcomes in sub-Saharan Africa, particularly in countries like Zimbabwe”.

6. REVIEWER COMMENT: Methods

This section could be structured a bit better to help with the flow. For example, in lines 130-131 under the authors just list “cross-sectional”. I encourage the authors to get rid of the study design heading and instead write a descriptive sentence that includes the study design but also adds some context about the study, including discussion of the data source, etc. and other important aspects that readers should be aware of.

AUTHOR RESPONSE: Thank you for the suggestion. We have revised to read: “This study employed a cross-sectional design to examine the integration of the status-neutral approach within existing HIV testing frameworks in sub-Saharan Africa. Data were sourced from a combination of health facility records, national health surveys, and interviews with healthcare providers. The cross-sectional nature of the study allowed us to capture a snapshot of the current state of HIV testing and treatment, providing valuable insights into the effectiveness and challenges of implementing the status-neutral approach alongside targeted testing methods”.

7. REVIEWER COMMENT: Authors should consider removing the sub-headings in line 133 and 138

AUTHOR RESPONSE: This suggestion has been implemented and the sub-headings removed.

8. REVIEWER COMMENT: In line 142 authors mention four subunits of ATP but name only three

AUTHOR RESPONSE: You may have missed the 4th, they are all mentioned as follows: “Four sub-units under ATP, namely, 1 HIV Prevention, 2Care and Treatment, 3Prevention of Mother to Child Transmission (PMTCT) and 4Monitoring and Evaluation (M & E)..

9. REVIEWER COMMENT: Description of study sites, lines 154-165, at times is unclear, authors should revise to improve readability. Also, the discussion of each province is unbalanced. I would also suggest the authors include data on the HIV prevalence and incidence rates in the four provinces. The authors discuss how these provinces were selected later on in the study, but that information should be moved to here to enhance comprehension

AUTHOR RESPONSE: Thank you for raising this important concern. Find below the revision: “The study sites included four provinces selected out of the ten provinces in Zimbabwe, based on their diverse geographical and demographic characteristics as well as varying levels of HIV prevalence and incidence. Manicaland Province: Located in eastern Zimbabwe, Manicaland is the second-most populous province after Harare, with a population of approximately 1.75 million as of the 2012 census. This province has been significantly impacted by the HIV epidemic, with high prevalence rates, making it a critical area for studying HIV interventions[18]. Mashonaland West Province: Situated to the north of Zimbabwe, Mashonaland West shares an international border with Zambia. It borders several other provinces internally, including Midlands, Matabeleland North, Mashonaland Central, Harare, and Mashonaland East. The region has a notable HIV prevalence, warranting focused efforts on HIV testing and treatment [19]. Matabeleland South Province: This province covers the southeastern plateau of Zimbabwe and stretches to the borders with Botswana and South Africa. The area is characterized by high HIV incidence, particularly in cross-border communities, highlighting the need for targeted HIV prevention and treatment strategies [20]. Midlands Province: Midlands province spans an area of 49,166 square kilometres and has a population of approximately 1.61 million. With its diverse population and significant HIV prevalence rates, this province provides a vital context for examining the effectiveness of HIV interventions[21].

These provinces were selected to provide a comprehensive understanding of the status-neutral approach and its integration into existing HIV testing frameworks across diverse settings. The varying HIV prevalence and incidence rates in these regions underscore the importance of context-specific strategies for HIV prevention and treatment”

10. REVIEWER COMMENT: Discussion of the sampling strategy should precede the discussion of the client population; authors should revise the order in which the sampling strategy and sample are discussed to improve comprehension

AUTHOR RESPONSE: Thank you for this observation, the sections have been correctly placed, sampling strategy first, then the client population.

11. REVIEWER COMMENT: In line 201 the authors states that no personal identifiers were collected but in line 199 it states that “name of patient” was collected from the report

AUTHOR RESPONSE: Thank you for this observation. We have revised as follows: “The following variables were collected: HTS number, name of the facility, age, sex, screening for an HIV test, reason for an HIV test, HIV test result, linkage to post-test services (yes/no), and specific services linked to. To ensure privacy, patient names were not included in the final dataset used for analysis. Therefore, no personal identifiers were collected in the analyzed data”

12. REVIEWER COMMENT: In line 217-219 authors explanation for why the study was exempt from ethical clearance makes it seem like DHIS2 database does not collect personally identifiable information, authors should revise justification and state that they received de-identified data if that was the case, and briefly explain the process

AUTHOR RESPONSE: We acknowledge your opinion. Here is the revision: “Our study was exempted from ethical clearance because it utilized routinely collected data extracted from the DHIS2 database. The data we received were de-identified, with all patient-identifying information removed prior to analysis. This ensured the privacy and confidentiality of the individuals' information. As a result, no primary data were collected, and the use of de-identified data qualified the study for ethical exemption.”

13. REVIEWER COMMENT: It would be helpful for the authors to briefly describe the complementary approach that fuses status neutral and targeted testing, is there a model that can be referenced? Both approaches are described but there is not a clear description of the kind of testing the authors seem to be advocating for.

AUTHOR RESPONSE: Thank you for pointing this out. Unfortunately there is no 3rd model that can be referenced. However, we have revised, under discussion intro to read: “A key finding in this study is that the status-neutral approach to HIV testing complements targeted HIV testing by creating a more inclusive and comprehensive testing framework. This dual strategy allows for prioritized testing of high-risk populations while ensuring that all individuals, regardless of their HIV status, have access to HIV prevention and treatment services. By integrating these approaches, we can enhance case identification, improve linkage to care, and ultimately reduce HIV transmission rates. This model aligns with the Comprehensive 95% targets set by UNAIDS, which emphasize the importance of linkage to comprehensive HIV prevention services.”

14. REVIEWER COMMENT: Authors should describe the variables, including what screening for an HIV test, linkage to post-test services, and specific services entail

AUTHOR RESPONSE: Thank you for pointing this out. We have added detail on the variables as follows: “Variables Collected: Screening for an HIV test: This variable refers to the initial assessment process conducted to determine an individual's risk of HIV infection and the need for an HIV test. It includes questions about sexual behavior, history of sexually transmitted infections, and other risk factors that might indicate the need for HIV testing. Linkage to post-test services: This variable indicates whether individuals who tested positive for HIV were connected to appropriate follow-up services. This includes referrals to healthcare facilities, counseling services, and support groups to ensure continuous engagement in care and treatment adherence.

Specific services linked to: These variable details the specific types of post-test services that individuals were connected to. These services may include antiretroviral therapy (ART), prevention of mother-to-child transmission (PMTCT) programs, pre-exposure prophylaxis (PrEP) for those at high risk, and other medical and social support services aimed at improving health outcomes for people living with HIV.”

15. VREVIEWER COMMENT: Discussion

The authors discuss their results and implications very succinctly. It would be helpful if the authors provided some discussion on specifically how this approach can be implemented and why, to illustrate the contribution of the findings of study and/or discussion of how the authors see their findings directly affecting health care service delivery.

AUTHOR RESPONSE: Thank you for the compliment and valuable feedback. We have added the following detail under Implications for policy and practice: “To effectively implement the status-neutral targeted HIV testing approach in Zimbabwe and beyond, several actionable steps and policy changes are necessary. Key stakeholders must also be identified to drive this initiative forward. Below are the recommended steps, policy adjustments, and potential barriers:

Actionable Steps:

i. Policy Development and Alignment: Develop national policies that support the integration of the status-neutral approach with targeted HIV testing. These policies should align with existing health strategies and frameworks to ensure seamless implementation.

ii. Stakeholder Engagement: Engage key stakeholders, including government health departments, non-governmental organizations (NGOs), community leaders, healthcare providers, and international partners such as the World Health Organization (WHO) and UNAIDS. Collaborative efforts are essential for resource mobilization, advocacy, and sustained support.

iii. Capacity Building: Invest in the training and capacity building of healthcare providers to enhance their understanding and implementation of the integrated approach. Training programs should focus on the benefits, procedures, and best practices for combining status-neutral and targeted testing methods.

iv. Resource Allocation: Ensure adequate resources are available for the implementation of the dual strategy. This includes funding for testing kits, infrastructure, personnel, and support services necessary for continuous engagement in care.

v. Community Outreach and Education: Launch community outreach programs to raise awareness about the integrated approach. Educational campaigns should aim to reduce stigma, encourage testing, and promote the benefits of continuous care and targeted interventions.

Policy Adjustments:

i. Regulatory Support: Implement regulatory frameworks that facilitate the integration of the status-neutral approach into existing HIV testing guidelines. This may involve revising existing protoc

---

## [Decision Letter · Decision Letter 2]

19 Jan 2025

PONE-D-24-14652R2Integrating Status-Neutral and Targeted HIV Testing in Zimbabwe: A Complementary StrategyPLOS ONE

Dear Dr. Mugauri,

Thank you for submitting your manuscript to PLOS ONE. After careful consideration, we feel that it has merit but does not fully meet PLOS ONE’s publication criteria as it currently stands. Therefore, we invite you to submit a revised version of the manuscript that addresses the points raised during the review process.

We look forward to receiving your revised manuscript.

Kind regards,

Zypher Jude G. Regencia, Ph.D.

Academic Editor

PLOS ONE

**Journal Requirements:**

Reviewers' comments:

Reviewer's Responses to Questions

**Comments to the Author**

1. If the authors have adequately addressed your comments raised in a previous round of review and you feel that this manuscript is now acceptable for publication, you may indicate that here to bypass the “Comments to the Author” section, enter your conflict of interest statement in the “Confidential to Editor” section, and submit your "Accept" recommendation.

Reviewer #3: All comments have been addressed

Reviewer #4: (No Response)

2. Is the manuscript technically sound, and do the data support the conclusions?

Reviewer #3: Yes

Reviewer #4: Yes

3. Has the statistical analysis been performed appropriately and rigorously? 

Reviewer #3: Yes

Reviewer #4: Yes

4. Have the authors made all data underlying the findings in their manuscript fully available?

Reviewer #3: Yes

Reviewer #4: Yes

5. Is the manuscript presented in an intelligible fashion and written in standard English?

Reviewer #3: Yes

Reviewer #4: Yes

6. Review Comments to the Author

**Reviewer #3: ** (No Response)

**Reviewer #4: ** Thank you for providing the opportunity to re-review the paper post revisions from the authors. I am satisfied with all the changes. One minor suggestion I have is that, in addition to the detail provided on the included provinces, I recommend that the authors to consider including the prevalence rates for each province into the manuscript.

7. PLOS authors have the option to publish the peer review history of their article (what does this mean? ). If published, this will include your full peer review and any attached files.

**Do you want your identity to be public for this peer review?** For information about this choice, including consent withdrawal, please see our Privacy Policy .

Reviewer #3: **Yes: ** Dr. Prachi Joshi

Reviewer #4: No

---

## [Author Response · Author response to Decision Letter 3]

20 Jan 2025

We thank you all for your time and dedication in going through this paper once more and we acknowledge the reviews and comments from the editor and reviewers. We provide a point-by-point response to each comment with each response introduced by the words “AUTHOR RESPONSE” in capital letters. We have revised the paper accordingly in response to these comments as indicated in our responses below and in tracked changes in the manuscript. We hope that the revised paper meets your expectations and may now be suitable for publication in your journal.

Yours sincerely,

Hamufare Mugauri

REVIEWER COMMENT: Thank you for providing the opportunity to re-review the paper post revisions from the authors. I am satisfied with all the changes. One minor suggestion I have is that, in addition to the detail provided on the included provinces, I recommend that the authors to consider including the prevalence rates for each province into the manuscript.

AUTHOR RESPONSE

Thank you for your appreciation and the comment. . We are pleased to hear that you are satisfied with the changes we have made. We appreciate your suggestion to include the prevalence rates for each province. We agree that this addition would enhance the depth and clarity of our manuscript. We have therefore incorporated the prevalence rates for Manicaland, Mashonaland West, Matabeleland South, and Midlands Province in the revised version of the paper as follows: “Manicaland Province: Located in eastern Zimbabwe, Manicaland is the second-most populous province after Harare, with a population of approximately 1.75 million as of the 2022 census. This province has been significantly impacted by the HIV epidemic, with a prevalence of 9.40% in 2024. . This makes it a critical area for studying HIV interventions[18, 19]. Mashonaland West Province: Situated to the north of Zimbabwe, Mashonaland West shares an international border with Zambia. It borders several other provinces internally, including Midlands, Matabeleland North, Mashonaland Central, Harare, and Mashonaland East. The region had a notable HIV prevalence of 9.60% in 2024, warranting focused efforts on HIV testing and treatment [19, 20]. Matabeleland South Province: This province covers the southeastern plateau of Zimbabwe and stretches to the borders with Botswana and South Africa. The area is characterized by high HIV incidence, particularly in cross-border communities, with a prevalence rate of 17.30% in 2024, highlighting the need for targeted HIV prevention and treatment strategies [19, 21]. Midlands Province: Midlands province spans an area of 49,166 square kilometres and has a population of approximately 1.61 million. With its diverse population and significant HIV prevalence rate of 10.94% in 2024, , this province provides a vital context for examining the effectiveness of HIV interventions[19, 22].”

---

## [Editor Report · Decision Letter 3]

3 Feb 2025

Integrating Status-Neutral and Targeted HIV Testing in Zimbabwe: A Complementary Strategy

PONE-D-24-14652R3

Dear Dr. Magauri,

We’re pleased to inform you that your manuscript has been judged scientifically suitable for publication and will be formally accepted for publication once it meets all outstanding technical requirements.

Kind regards,

Zypher Jude G. Regencia, Ph.D.

Academic Editor

PLOS ONE
---

## [Editor Report · Acceptance letter]

PONE-D-24-14652R3

PLOS ONE

Dear Dr. Mugauri,

I'm pleased to inform you that your manuscript has been deemed suitable for publication in PLOS ONE. Congratulations! Your manuscript is now being handed over to our production team.

Kind regards,

on behalf of

Dr. Zypher Jude G. Regencia

Academic Editor

PLOS ONE